# Prevalence and Distribution of MUTYH Pathogenic Variants, Is There a Relation with an Increased Risk of Breast Cancer?

**DOI:** 10.3390/cancers16020315

**Published:** 2024-01-11

**Authors:** Jesús Peña-López, Diego Jiménez-Bou, Icíar Ruíz-Gutiérrez, Gema Martín-Montalvo, María Alameda-Guijarro, Antonio Rueda-Lara, Leticia Ruíz-Giménez, Oliver Higuera-Gómez, Alejandro Gallego, Ana Pertejo-Fernández, Darío Sánchez-Cabrero, Jaime Feliu, Nuria Rodríguez-Salas

**Affiliations:** 1Department of Medical Oncology, Hospital Universitario La Paz, 28046 Madrid, Spain; 2Department of Medical Oncology, Clínica Universidad de Navarra, 28027 Madrid, Spain

**Keywords:** MUTYH, breast cancer, hereditary, polyposis

## Abstract

**Simple Summary:**

Colorectal cancer is often associated with *MUTYH* mutations, but their connection to breast cancer remains unclear. We aimed to assess if *MUTYH* mutations contribute to breast cancer susceptibility. Analyzing data from 3598 patients, we found *MUTYH* mutations in 1.6%, with a significant association in colonic polyposis cases. However, our findings did not reveal a substantial association between *MUTYH* mutations and breast cancer. These insights emphasize the need for cautious interpretation of *MUTYH* mutations in breast cancer risk assessments.

**Abstract:**

Background: *MUTYH* has been implicated in hereditary colonic polyposis and colorectal carcinoma. However, there are conflicting data refgarding its relationship to hereditary breast cancer. Therefore, we aimed to assess if *MUTYH* mutations contribute to breast cancer susceptibility. Methods: We retrospectively reviewed 3598 patients evaluated from June 2018 to June 2023 at the Hereditary Cancer Unit of La Paz University Hospital, focusing on those with detected *MUTYH* variants. Results: Variants of *MUTYH* were detected in 56 patients (1.6%, 95%CI: 1.2–2.0). Of the 766 patients with breast cancer, 14 patients were carriers of *MUTYH* mutations (1.8%, 95%CI: 0.5–3.0). The prevalence of *MUTYH* mutation was significantly higher in the subpopulation with colonic polyposis (11.3% vs. 1.1%, *p* < 0.00001, OR = 11.2, 95%CI: 6.2–22.3). However, there was no significant difference in the prevalence within the subpopulation with breast cancer (1.8% vs. 1.5%, *p* = 0.49, OR = 1.2, 95%CI: 0.7–2.3). Conclusion: In our population, we could not establish a relationship between *MUTYH* and breast cancer. These findings highlight the necessity for a careful interpretation when assessing the role of *MUTYH* mutations in breast cancer risk.

## 1. Introduction

Colorectal cancer (CRC) is the third most common cancer and the second leading cause of cancer death worldwide [1]. Pathogenic genetic mutations in genes associated with a significant or moderate cancer risk are identified in 6–10% of all CRC and even in 20% of cases diagnosed before the age of 50. Hereditary colorectal cancer syndrome is classified as non-polyposis and polyposis. The most common is hereditary non-polyposis colorectal cancer (HNPCC), also known as Lynch syndrome, which accounts for about 3% of cases. In the case of hereditary polyposis colorectal cancer syndrome, the most frequent syndrome is familial adenomatous polyposis (FAP) caused by mutation in the adenomatous polyposis coli (*APC*) gene, which accounts for about 1% of cases. There are also other polyposis syndromes, such as *MUTYH*-associated polyposis (MAP), which accounts for about 0.5% of cases [2]. MAP is an autosomal recessive syndrome caused by biallelic germline mutations (either homozygous or compound heterozygous) on *MUTYH* that predisposes to both colonic polyposis and colorectal carcinoma [3,4].

*MUTYH* is a gene located in the locus 1p34.1 [5]. It encodes a DNA glycosylase of the base excision repair (BER) that prevents mutations due to oxidative DNA damage. Specifically, it targets guanine (G): cytosine (C) → thymine (T): adenosine (A) transversions mediated by guanine oxidation (8-oxoG) that mispairs with A residues instead of C. *MUTYH*-inherited variants can induce somatic APC mutations in colorectal tumors [6]. The two most common pathogenic/like pathogenic *MUTYH* mutations in the Caucasian population are p.Y179C (also known as Y151C, Y165C or Y176C) and p.G382D (also known as G368D or G393D) [7,8,9]. Although less described, these mutations are also linked to the risk of extracolonic cancers like duodenal, ovarian, bladder or skin [10]. 

There has been controversy about whether *MUTYH* mutations are associated with breast cancer. Breast cancer (BC) is the most common cancer and the fourth leading cause of cancer death worldwide [1]. Between 5% and 10% of individuals with BC have a genetic predisposition. Most cases of hereditary breast and ovarian cancer syndrome (HBOCS) are caused by germline mutations in *BRCA1* or *BRCA2*. This syndrome is characterized by an increased risk of certain types of cancers, especially breast cancer, and an early onset of these cancers. However, this syndrome can occur without *BRCA1/2* mutation due to alteration of other genes involved in DNA repair like *RAD51*, *ATM*, *CHEK2* or *BRIP1* [11].

The first studies that suggested the correlation between *MUTYH* and BC appeared in the year 2005. Nielsen et al. 2005 detected *MUTYH* pathogenic variations in 40 patients out of 170 patients with colonic polyposis and an absence of the *APC* mutation. In that group, 18% of female patients (4/22) presented with breast cancer [12]. There are many descriptive studies detecting *MUTYH* mutations in breast cancer patients [13]. Moreover, there is a molecular plausibility for that correlation. BRCA1 and BRCA2 are two tumor suppressor genes that participate in the repair of double-stranded DNA breaks through the homologous recombination pathway [11]. BRCA has also been linked to the repair processes of the oxidative lesion 8-oxoG, where MUTYH also performs its function [14].

However, there are case–control studies that have found no correlation between carriers of *MUTYH* mutations and breast cancer [15]. Several studies have examined the frequency of the two common missense mutations (p.Y179C and p.G382D), and approximately 1–2% of the general population (of European origin) is predicted to be a carrier [16]. So, it is reasonable that these variants may appear as incidental findings in germline studies. 

The objective of this study is to describe the patients carrying *MUTYH* mutations in our center and to try to determine if there is a relationship with breast cancer. Elucidating this relationship has implications for diagnosis (need to include *MUTYH* in gene panels to study HBOCS), screening (need for breast imaging studies in *MUTYH* carriers) and patient information.

## 2. Materials and Methods

A retrospective study was conducted at the ‘Hereditary Cancer Unit’ of La Paz University Hospital (Madrid) from June 2018 to June 2023. The study included the adult population (>18 years old) which attended during this period. The demographic, clinical and familial histories were collected from genealogical trees and medical records. We selected patients in whom a variant was detected in *MUTYH*. 

The indication for genetic testing in suspected hereditary syndrome was performed following the NCCN (National Comprehensive Cancer Network) guideline criteria. For the genetic study of *MUTYH*, DNA was extracted from peripheral blood using a magnetic separation method on the Chemagic 360 or Chemagen Magtration system 8Lx (PerkinElmer). Then, the study could be performed in two settings. If a familial genetic variant in the index case was already identified, cascade testing was conducted for at-risk relatives. This involved amplification through polymerase chain reaction (PCR), followed by sequencing of the amplified fragments using capillary electrophoresis on an ABI3730 (Applied Biosystems). If hereditary syndrome was suspected, Target Next Generation Sequencing (NGS) panel based in hybridization capture was carried out. It was performed on the MiSeq, HiSeq or NextSeq platform (Illumina) following the paired-ends strategy. Selection of target regions was performed using OncoClever-GeneSGKit capture. The bioinformatics analysis (comparison of the DNA sequence obtained with the reference genomic sequence, GRCh38) was conducted with platforms like DATA GENOMICS (IMEGEN) or SOPHIA-GENETICS (SOPHIA DDM^®^). If FAP was suspected, the NGS included the genes *APC* and *MUTYH*. In patients with early onset colorectal cancer, the NGS included the genes *MLH1*, *MSH2*, *MSH6*, *PMS2*, *EPCAM*, *POLE*, *POLD*, *MUTYH* and *APC*. If HBOCS was suspected, the NGS included *MUTYH*, *BRCA1*, *BRCA2*, *PTEN*, *CDH1*, *CHEK2*, *BLM*, *XRCC2*, *MSH2*, *MSH6*, *MLH1*, *FAM175A*, *ATM*, *PALB2*, *STK11*, *MEN1*, *BARD1*, *BRIP1*, *RAD51C*, *RAD51D*, *TP53*, *MRE11A*, *RAD50*, *NBN*, *PMS2* and *EPCAM*. Any changes detected by NGS are subsequently confirmed using automatic bidirectional sequencing (Sanger). The variants were analyzed using databases such as ClinVar, HGMD Pro or LOVD. The nomenclature used was approved by the Human Genome Variation Society (HGVS). The variants were classified in accordance with the American College of Medical Genetics and Genomics (ACMG) into a five-tier system (class 5: deleterious; class 4: likely deleterious; class 3: uncertain significance; class 2: likely benign; and class 1: benign).

The prevalence of the mutations detected and patient characteristics were reported with descriptive statistics. The demographic, clinical and pathological characteristics were compared using chi-square tests for categorical variables. The odds ratio (OR) was compared between the different groups for each clinical factor using Fisher’s exact test. *p* values <0.05 were considered statistically significant. Statistical analysis was carried out using IBM SPSS Statistics for Windows Version 19.0 (IBM Corporation, Armonk, NY, USA).

## 3. Results

From June 2018 to June 2023, 3598 patients were assessed in the Hereditary Cancer Unit. The median age was 56 years old (range: 18–97), and the majority were female (*n* = 2690, 74.8%). In relation to the most frequent underlying pathology, 766 had breast cancer (21.3%), 399 had colorectal cancer (11.1%), 183 had ovarian cancer (5.1%), 175 had pancreatic cancer (4.9%), 150 had colonic polyposis (4.2%) and 85 had prostate cancer (2.4%). Of the 766 breast cancer patients, 13 were male (1.7%).

### 3.1. Characteristics of MUTYHmut Carriers

Variants of *MUTYH* were detected in 56 patients (1.6%, 95%CI: 1.2–2.0). The median age was 60 years old (range: 18–85), and the majority were female (*n* = 44, 78.6%). In total, 36 patients were the index case (64.3%). The suspected hereditary syndrome that prompted the study were HBOCS (*n* = 27, 48.2%), familial polyposis (*n* = 18, 32.1%), HNPCC (*n* = 7, 12.5%), hereditary leukemia (*n* = 3, 5.4%) and hereditary renal cancer (*n* = 1, 1.8%). The underlying pathology was colonic polyposis (*n* = 17, 30.4%), breast cancer (*n* = 14, 25%), colorectal cancer (*n* = 7, 12.5%), ovarian cancer (*n* = 5, 8.9%), pancreatic cancer (*n* = 3, 5.4%), renal cancer (*n* = 3, 5.4%), prostate cancer (*n* = 2, 3.6%), hematological cancer (*n* = 2, 3.6%), melanoma (*n* = 2, 1.8%), endometrial cancer (*n* = 1, 1.8%) and lung cancer (*n* = 1, 1.8%). A total of 10 patients had multiple cancers (17.8%), 15 patients did not have cancer or polyposis (26.8%), and 3 patients had colonic polyps and cancer (endometrial, ovarian and colorectal). Patient characteristics are summarized in Table 1.

### 3.2. Genetic Findings of MUTYHmut Carriers

The most frequent mutations were p.G382D/Class 5 (*n* = 39, 62.9%), p.Y179C/Class 5 (*n* = 6, 9.7%), p.E410Gfs*43/Class 5 (*n* = 3, 4.8%), p.R368Qfs*164/Class 4 (*n* = 3, 4.8%), p.Q338*/Class 5 (*n* = 2, 3.2%), p.R97Q/Class 3 (*n* = 1, 1.6%), p.R109W/Class 4 (*n* = 1, 1.6%), p.E453del/Class 5 (*n* = 1, 1.6%), p.D147H/Class 3 (*n* = 1, 1.6%), p.Q324=/Class 1 (*n* = 1, 1.6%), p.R426C/Class 3 (*n* = 1, 1.6%), c.933+3A>C/Class 3 (*n* = 1, 1.6%), c.934-2A>G/Class 4 (*n* = 1, 1.6%) and c.997+1G>T/Class 4 (*n* = 1, 1.6%). The majority were heterozygous (*n* = 50, 89.2%). Two patients were p.G382D homozygous (one patient with colonic polyps and another with sister with colonic polyps). Four patients were compound heterozygous: p.G382D + p.R97Q (one patient with breast cancer), p.G382D + p.R109W (one patient with colonic polyps), p.G382D + p.E453del (one patient with endometrial cancer), and p.Y179C + p.E410Gfs*43 (one patient with nonpolyposis colorectal cancer). 

In five patients, there was a pathogenic/likely pathogenic mutation in another gene: *RAD50* p.E723Gfs*5/Class 5 (one patient with breast cancer), *RAD50* p.E723Gfs*5/Class 5 (one patient with nonpolyposis colorectal cancer), *APC* p.N1739fs/Class 5 (one patient with polyposis), *NF1* p.N2220Ifs*25/Class 5 (one patient with family history of breast cancer) and *SDHB* p.P56Yfs*5/Class 5 (one patient with renal cancer). In another four patients, there was a variant of uncertain significance: *APC* p.D1711V/Class 3 (one patient with polyposis), *BRCA2* p.E170A/Class 3 (one patient with breast cancer and nonpolyposis colorectal cancer), *BRCA2* Class 3 (one patient with ovarian cancer) and ATM c.720T>C/Class3 (one patient with breast cancer). 

Genetic results are summarized in Table 2. Table 3 summarizes the phenotype mutation in MUTYH and the presence of other mutations in other genes for each patient.

### 3.3. Breast Cancer in MUTYHmut Carriers

In relation to patients with breast cancer with MUTYH mutations (*n* = 14). The median age was 51.5 years old (range: 33–78), and all were females. The majority were ductal (*n* = 11, 78.6%), histological grade 2 (*n* = 7, 50%) and stage I-II (*n* = 10. 71.4%). The molecular subtypes were luminal A (*n* = 5, 35.7%), luminal B (*n* = 5, 35.7%) and triple negative (*n* = 4, 28.6%). The *MUTYH* mutations found were p.G382D (*n* = 7, one compound heterozygous with p.R97Q), p.Y179C (*n* = 2, one with ATM mutation), p.Q338* (*n* = 2), p.E410Gfs*43 (*n* = 1, with BRCA2 mutation), c.933+3A>C (*n* = 1) and c.934-2A>G (*n* = 1). In four patients with polyposis and without breast cancer, there was a family history of breast cancer.

### 3.4. Prevalence of MUTYH Mutations and Association with Pathologies

The prevalence of *MUTYH* mutations was 1.6% (95%CI: 1.2–2.0) in our setting, 11.3% (95%CI: 6.3–16.4) in colonic polyposis, 1.7% (95%CI: 0.0–3.6) in pancreatic cancer, 2.7% (95%CI: 0.4–5.0) in ovarian cancer, 2.4% (2/85, 95%CI 0.0–5.6) in prostate cancer, 1.8% (95%CI: 0.9–2.8) in breast cancer and 1.8% (95%CI: 0.5–3.0) in colorectal cancer. The prevalence of *MUTYH* mutation was significantly higher in the subpopulation with polyposis vs. without (11.3% vs. 1.1%, *p* < 0.00001) and was not significantly different in the subpopulation with colorectal cancer vs. without (1.8% vs. 1.6%, *p* = 0.74) or breast cancer vs. without (1.8% vs. 1.5%, *p* = 0.49). The odds ratio (OR) for the presence of *MUTYH* mutation was only significant in colonic polyposis (OR = 11.2, 95%CI: 6.2–22.3). It was not significant in the case of ovarian cancer (OR = 1.8, 95%CI: 0.7–4.7), prostate cancer (OR = 1.5, 95%CI: 0.4–6.4), breast cancer (OR = 1.2, 95%CI: 0.7–2.3), colorectal cancer (OR = 1.1, 95%CI: 0.5–2.6) or pancreatic cancer (OR = 1.1, 95%CI: 0.3–3.6). The results are summarized in Table 4.

## 4. Discussion

It is necessary to know the characteristics of *MUTYH* carriers in order to provide the best prognostic information to our patients. The results of our study indicate that the prevalence of germline alterations in *MUTYH* in our series is 1.6%, and among breast cancer patients, it is 1.8%. A statistically significant association was detected between *MUTYH* carriers and the presence of colonic polyposis with an OR of 11.2. It was not possible to detect an association between *MUTYH* mutations and breast cancer, obtaining a non-significant OR. In relation to other types of cancer, no significant association with heterozygous *MUTYH* mutations was found. Therefore, it cannot be affirmed that there is a relationship between the presence of *MUTYH* and breast cancer, one of the objectives of this study.

Similar to previous studies, in our setting, we have obtained a prevalence of *MUTYH* carriers of around 1–2%, with a predominance of G382D and Y179C mutations [16]. The prevalence of *MUTYH* in other studies has obtained values between 0.3 and 5.6% as illustrated in Table 5. Particularly interesting is the study of Kurian et al. 2021 with a sample size of about 15,000 patients who obtained a value of 1.4%. Therefore, our value of 1.8% is consistent with previous evidence.

Former studies of the association between *MUTYH* variants and the risk of breast cancer have yielded conflicting results as shown in Table 6. However, studies with larger sample sizes do not seem to detect a relationship. This is consistent with the data from our work.

Interestingly, our study does not detect a relationship between *MUTYH* mutations and colorectal cancer. Nevertheless, there are studies that describe it. For example, Win et al. 2011, in monoallelic carriers, reported an elevated risk of colorectal cancer (OR 2.0), gastric cancer (OR 3.4), liver cancer (OR 3.1) and endometrial cancer (OR 2.3) but not in cancers of the breast, lung, kidney, prostate, pancreas or brain [33]. Win et al. 2014 described a slightly increased risk of colorectal cancer in patients with heterozygous *MUTYH* mutation, especially if there is a family history [39]. However, other studies have found no association between colon cancer and heterozygous mutations in *MUTYH* [40]. 

On the other hand, alterations in *MUTYH* are indeed related to colonic polyposis. But this usually occurs when the mutation is homozygous. We have obtained a significantly higher prevalence of heterozygous *MUTYH* mutations in patients with colonic polyposis and a significant and large OR of 11.2. Croitoru et al. 2004 showed that loss of heterozygosity phenomena could occur in patients with heterozygous *MUTYH* germline mutations and colonic polyps [41].

A strength of this study is that it is able to gather fourteen patients with breast cancer and *MUTYH* mutation. Most descriptive studies of hereditary breast cancer do not focus on *MUTYH,* but rather, it is a finding in the context of a panel of genes. Therefore, in these studies, they generally obtain fewer than 10 patients with breast cancer and *MUTYH* mutation. 

There are several limitations. First, it is a single-center study, and *MUTYH* was studied with different methods in each patient. The choice of genes included in the gene panel varies depending on the diagnostic suspicion, thereby introducing a potential confounding variable in the extrapolation of our results. Moreover, the most frequent reason for detecting *MUTYH* alterations was in the context of a suspected HBOCS. This might be attributed to its prevalence as the primary motive for assessment within the Hereditary Cancer Unit. Another important aspect is to acknowledge the inherent ascertainment biases in a retrospective observational study in which the population is chosen specifically for genetic testing related to cancer susceptibility.

## 5. Conclusions

In summary, these data do not support a clinically significant association of breast cancer risk with monoallelic *MUTYH* carrier status. While multigene panel testing yields extensive and practical datasets, it is crucial to exercise caution when interpreting this information on an individual basis. This approach helps prevent the unintentional dissemination of potentially misleading clinical information that could harm patients. Additionally, when dealing with heterozygous mutations in the *MUTYH* gene, it is essential to interpret them with care, given their relatively frequent occurrence in the general population and the evolving insights into their significance.

## Figures and Tables

**Table 1 cancers-16-00315-t001:** Baseline characteristics. HBOCS = hereditary breast and ovarian cancer syndrome; HNPCC = hereditary non-polyposis colorectal cancer.

Characteristics	N = 56
Age (years)—Median (range)	60 (18–85)
Sex—*n* (%) -Male -Female	12 (21.4%) 44 (78.6%)
Index case—*n* (%) -Yes -No	36 (64.3%) 20 (35.7%)
Suspected hereditary syndrome—*n* (%) -HBOCS -Familial polyposis -HNPCC-Hereditary leukemia -Hereditary renal cancer	27 (48.2%) 18 (32.1%) 7 (12.5%) 3 (5.4%) 1 (1.8%)
Pathology—*n* (%) -Non cancer/polyposis -Colonic polyposis -Breast cancer -Colorectal cancer -Ovarian cancer -Pancreatic cancer -Renal cancer -Prostate cancer -Hematological cancer -Melanoma -Endometrial cancer -Lung cancer	15 (26.8%) 17 (30.4%) 14 (25%) 7 (12.5%) 5 (8.9%) 3 (5.4%) 3 (5.4%) 2 (3.6%) 2 (3.6%) 2 (3.6%) 1 (1.8%) 1 (1.8%)

**Table 2 cancers-16-00315-t002:** Genetic results.

Characteristics	N = 56
*MUTYH* allele status—*n* (%) -Homozygous -Compound heterozygous -Heterozygous	2 (3.6%) 4 (7.1%) 50 (89.2%)
*MUTYH* mutations—*n* (%) -p.G382D/Class 5 -p.Y179C/Class 5 -p.E410Gfs*43/Class 5 -p.R368Qfs*164/Class 4 -p.Q338*/Class 5 -p.R97Q/Class 3 -p.R109W/Class 4 -p.E453del/Class 5 -p.D147H/Class 3-p.Q324=/Class 1 -p.R426C/Class 3 -c.933+3A>C/Class 3 -c.934-2A>G/Class 4 -c.997+1G>T/Class 4	62 (100%) 39 (62.9%) 6 (9.7%) 3 (4.8%) 3 (4.8%) 2 (3.2%) 1 (1.6%) 1 (1.6%) 1 (1.6%) 1 (1.6%) 1 (1.6%) 1 (1.6%) 1 (1.6%) 1 (1.6%) 1 (1.6%)
Other mutations—*n* -*BRCA2* -*RAD50* -*APC* -*ATM* -*NF1* -*SDHB*	2 (3.6%) 2 (3.6%) 2 (3.6%) 1 (1.8%) 1 (1.8%) 1 (1.8%)

**Table 3 cancers-16-00315-t003:** Summary of patients with *MUTYH* mutations. The classification of the mutation according to ACMG is indicated in parenthesis. HBOCS = hereditary breast and ovarian cancer syndrome, HNPCC = hereditary non-polyposis colorectal cancer, HL = hereditary leukemia, HRC = hereditary renal cancer, Y = yes, N = no, BC = breast cancer, OC = ovarian cancer, PCC = pancreatic cancer, PC = prostate cancer, CRC = colorectal cancer, RC = renal cancer, EC = endometrial cancer, LC = lung cancer, L = leukemia, M = melanoma.

Nº	Sex/ Age	Suspected Syndrome	Index Case	Polyposis	Tumor	*MUTYH* Mutation	Other Mutation
1	F35	HBOCS	Y	N	BC + RC	p.G382D (5) heterozygosis	
2	F33	HBOCS	Y	N	BC + OC	p.G382D (5) heterozygosis	
3	F33	HBOCS	Y	N	BC	p.G382D (5) heterozygosis	
4	F35	HBOCS	Y	N	BC	p.G382D (5) + p.R97Q (3) compound heterozygosis	
5	F38	HBOCS	Y	N	BC	p.G382D (5) heterozygosis	
6	F70	HBOCS	Y	N	BC	p.G382D (5) heterozygosis	
7	F62	HBOCS	Y	N	BC + PCC	p.G382D (5) heterozygosis	
8	F39	HBOCS	Y	N	BC	p.Y179C (5) heterozygosis	ATM c.720T>C (3)
9	F60	HBOCS	Y	N	BC + RC	p.Y179C (5) heterozygosis	
10	F43	HBOCS	Y	N	BC	p.Q338* (5) heterozygosis	
11	F60	HBOCS	Y	N	BC	p.Q338* (5) heterozygosis	
12	F60	HBOCS	Y	N	BC	c.934-2A>G (4) heterozygosis	
13	F76	HBOCS	Y	N	BC + LC	c.933+3A>C (3) heterozygosis	
14	F78	HNPCC	Y	N	BC + CRC + OC	p.E410Gfs*43 (5) heterozygosis	BRCA2 p.E170A (3)
15	F64	FAP	N	N	-	p.G382D (5) homozygosis	
16	F35	FAP	Y	Y	-	p.G382D (5) homozygosis	
17	F57	FAP	Y	Y	-	p.G382D (5) heterozygosis	APC p.N1739fs (5)
18	F56	FAP	Y	Y	-	p.G382D (5) heterozygosis	
19	F60	FAP	Y	Y	-	p.G382D (5) heterozygosis	APC p.D1711V (3)
20	M37	FAP	N	Y	-	p.G382D (5) heterozygosis	
21	F62	FAP	N	Y	-	p.G382D (5) heterozygosis	
22	M29	FAP	N	Y	-	p.G382D (5) heterozygosis	
23	F61	FAP	Y	Y	-	p.G382D (5) + p.R109W (4) compound heterozygosis	
24	F67	FAP	N	Y	-	p.G382D (5) heterozygosis	
25	M60	FAP	N	Y	-	p.G382D (5) heterozygosis	
26	F53	FAP	Y	Y	EC	p.G382D (5) + p.E453del (5) compound heterozygosis	
27	F76	FAP	N	N	-	p.Y179C (5) heterozygosis	
28	M85	FAP	Y	Y	CRC	p.Y179C (5) heterozygosis	
29	F51	FAP	Y	Y	-	p.Q324= (1) heterozygosis	
30	M76	FAP	Y	Y	-	p.D147H (3) heterozygosis	
31	M62	FAP	N	Y	-	p.R426C (3) heterozygosis	
32	M77	FAP	Y	Y	-	p.E410Gfs*43 (5) heterozygosis	
33	F24	HBOCS	N	N	-	p.G382D (5) heterozygosis	NF1 p.N2220Ifs*25 (5)
34	F59	HBOCS	Y	N	CRC + OC	p.G382D (5) heterozygosis	
35	F66	HBOCS	N	N	-	p.G382D (5) heterozygosis	
36	F69	HBOCS	N	N	-	p.G382D (5) heterozygosis	
37	M67	HBOCS	Y	N	M	p.G382D (5) heterozygosis	
38	F62	HBOCS	N	N	-	p.G382D (5) heterozygosis	
39	F54	HBOCS	N	N	-	p.G382D (5) heterozygosis	
40	F69	HBOCS	Y	N	CRC + OC	p.G382D (5) heterozygosis	BRCA2 (3)
41	F67	HBOCS	Y	Y	OC	p.G382D (5) heterozygosis	
42	M70	HBOCS	Y	N	PC + L	p.G382D (5) heterozygosis	
43	F37	HBOCS	N	N	-	p.G382D (5) heterozygosis	
44	F44	HBOCS	N	N	-	p.G382D (5) heterozygosis	
45	M63	HBOCS	Y	N	PC + PCC	p.Y179C (5) heterozygosis	
46	F69	HBOCS	Y	N	PCC	c.997+1G>T (4) heterozygous	
47	F50	HNPCC	Y	N	CRC	p.Y179C (5) + p.E410Gfs*43 (5) compound heterozygosis	RAD50 p.E723Gfs*5 (5)
48	F63	HNPCC	Y	N	CRC	p.G382D (5) heterozygosis	RAD50 p.E723Gfs*5 (5)
49	F50	HNPCC	N	N	-	p.G382D (5) heterozygosis	
50	F76	HNPCC	N	N	-	p.R368Qfs*164 (4) heterozygosis	
51	F39	HNPCC	N	N	-	p.R368Qfs*164 (4) heterozygosis	
52	F52	HNPCC	N	N	-	p.R368Qfs*164 (4) heterozygosis	
53	F73	HL	Y	N	L	p.G382D (5) heterozygosis	
54	F48	HL	N	N	-	p.G382D (5) heterozygosis	
55	M18	HL	N	N	-	p.G382D (5) heterozygosis	
56	M56	HRC	Y	N	RC + CRC + M	p.G382D (5) heterozygosis	SDHB p.P56Yfs*5 (5)

**Table 4 cancers-16-00315-t004:** Prevalence and odds ratio (OR) of *MUTYH* mutations in general population and across different types of tumors.

	Prevalence	OR
Our setting	1.6% (95%CI: 1.2–2.0)	-
Colonic polyposis	11.3% (95%CI: 6.3–16.4)	11.2 (95%CI: 6.2–22.3)
Ovarian cancer	2.7% (95%CI: 0.4–5.0)	1.8 (95%CI: 0.7–4.7)
Prostate cancer	2.4% (95%CI 0.0–5.6)	1.5 (95%CI: 0.4–6.4)
Breast cancer	1.8% (95%CI: 0.9–2.8)	1.2 (95%CI: 0.7–2.3)
Colorectal cancer	1.8% (95%CI: 0.5–3.0)	1.1 (95%CI: 0.5–2.6)
Pancreatic cancer	1.7% (95%CI: 0.0–3.6)	1.1 (95%CI: 0.3–3.6)

**Table 5 cancers-16-00315-t005:** Some descriptive studies describing the prevalence of *MUTYH* alterations in breast cancer patients. BC = breast cancer, OC = ovarian cancer, *BRCA*x = without *BRCA* mutation, HBOCS = hereditary breast and ovarian cancer syndrome, EOBC = early onset breast cancer.

Article	Population	Sample Size	Prevalence
Kurian et al. 2014 [17]	Females with BC	174	2.9%
Maxwell et al. 2015 [13]	Females with BC and *BRCA*x	278	2.6%
Ellingson et al. 2015 [18]	Females with BC	124	1.6%
Lin et al. 2016 [19]	Females with BC and HBOCS criteria	133	0.8%
Rummel et al. 2017 [20]	Females with EOBC	119	1.7%
Kaur et al. 2018 [21]	Females with BC	296	5.6%
Meiss et al. 2018 [22]	Females with BC	612	0.3%
Oliver et al. 2019 [23]	Latin Americans with HBOCS criteria	222	1.4%
Rizzolo et al. 2018 [24]	Italian males with BC	503	0.4%
Schneider et al. 2019 [25]	Patients with BC	146	0.7%
Ryu et al. 2020 [26]	Patients with BC and HBOCS criteria	507	0.6%
Chen et al. 2020 [27]	Chinese patients with BC	524	0.8%
Kurian et al. 2021 [28]	Patients with BC	15,256	1.4%
Oliveira et al. 2022 [29]	Patients with BC or OC	971	2.7%
Tatineni et al. 2022 [30]	Patients with BC and HBOCS criteria	922	1.4%
Our series	Patients with BC	766	1.8%

**Table 6 cancers-16-00315-t006:** Some case–control studies describing the association of *MUTYH* alterations with breast cancer patients. BC = breast cancer, *BRCA*x = without *BRCA* mutation, *MUTYH*mut = *MUTYH* mutation, P/LP = pathogenic/likely pathogenic, OR = odds ratio, SIR = standardized incidence ratio.

Article	Population	Results
Rennert et al. 2012 [31]	Sephardi patients with BC (*n* = 389) vs. controls (*n* = 541)	Positive. Increase in BC in patients with heterozygous P/LP *MUTYH* G382D (6.7% vs. 3.7%, OR 1.86, 95%CI 1.02–3.39; *p* = 0.04).
Rizzolo et al. 2018 [24]	Males with BC and *BRCA*x (*n* = 503) vs. controls (*n* = 1540)	Positive. Heterozygous Y165C was associated with increased BC (OR 4.54, 95%CI 1.17–17.58, *p* = 0.028).
Beiner et al. 2008 [32]	Patients with BC (*n* = 691) vs. controls (*n* = 812)	Negative. No association in BC risk and heterozygous *MUTYH*mut.
Win et al. 2011 [33]	First and second-degree relatives (*n* = 2179) of 144 incident CCR cases who were mono/bi-allelic *MUTYH*mut carriers vs. expected number of cancers in general population	Negative. No association in BC and *MUTYH*mut (SIR 1.27, 95%CI 0.84–1.99, *p* = 0.28).
Out et al. 2012 [34]	Patients with incident BC (*n* = 1469) and patients with BC and *BRCA*x (*n* = 471) vs. controls (*n* = 1666)	Negative. No association in BC and monoallelic *MUTYH*mut.
Win et al. 2016 [35]	First and second-degree relatives (*n* = 5158) of 266 probands with *MUTYH*mut (42 biallelic and 225 monoallelic) vs. expected number of cancers in general population	Negative. No association in BC and monoallelic *MUTYH*mut (HR 1.4, 95%CI 1.0–2.0).Nevertheless, in the subgroup of female monoallelic mutation carriers, there is an estimated cumulative risk to age 70 years of BC of 11% (95%CI, 8%–16%)
Kurian et al. 2017 [28]	Patients with BC (*n* = 26,384) vs. controls (*n* = 64,649)	Negative. No association in BC and biallelic *MUTYH*mut (OR 0.55, 95%CI 0.22–1.38, *p* = 0.2)
Jian et al. 2017 [36]	Chinese patients with BC (*n* = 120) vs. Chinese women with high-risk for BC (*n* = 120)	Negative. No association in BC and *MUTYH*mut (1.7% vs. 5.8%, *p* = 0.17).
Fulk et al. 2019 [15]	Females with BC (*n* = 30,456) vs. controls (*n* = 12,289)	Negative. No association in BC and monoallelic *MUTYH*mut (OR 1.01, 95%CI 0.85–1.21, *p* = 0.89).
Thompson et al. 2022 [37]	Females with BC (n = 20,043) vs. controls (*n* = 22,150)	Negative. No association in BC and monoallelic P/LP *MUTYH*mut (1.9% vs. 1.7%, OR 1.1, 95%CI 0.96–1.3, *p* = 0.15).
Guindalini et al. 2022 [38]	Brazilian patients with BC (*n* = 1663) vs. controls (*n* = 18,919)	Negative. No association in BC and monoallelic *MUTYH*mut G382D (1.2% vs. 0.9%, OR 1.4, 95%CI: 0.8–2.4; *p* = 0.29) or Y165C (0.8% vs. 0.4%, OR 1.9. 95%CI 0.9–3.9, *p* = 0.09).

## Data Availability

De-identified individual data might be made available following publication by reasonable request to the corresponding author.

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
