# Peer review of "Prevalence and Distribution of MUTYH Pathogenic Variants, Is There a Relation with an Increased Risk of Breast Cancer?"

_cancers, 2024, doi:10.3390/cancers16020315_

Round 1
Reviewer 1 Report
Comments and Suggestions for Authors
This is a well-designed and executed study of the MUTYH gene in breast cancer. The authors analyzed data from 3598 patients and, as expected, observed a significantly higher prevalence of MUTYH mutations in patients with colonic polyposis. Of the 766 patients with breast cancer, 14 patients were carriers of MUTYH mutations (1.8%, 95%CI: 0.5-3.0). The prevalence of MUTYH mutation was not significantly different in patients with breast cancer (1.8% vs 1.5%, p=0.49, OR=1.2, 95%CI: 0.7-2.3). The authors draw the valid conclusion that there is no relationship between MUTYH and breast cancer. The references are up-to-date.
Author Response
The interpretation of our work that you reflect in your description is what we intend to convey. Thank you very much for taking the time to review this manuscript.
Reviewer 2 Report
Comments and Suggestions for Authors
In this study, the authors aimed to investigate the role of MUTYH mutations in breast cancer susceptibility. They analyzed data from 3598 patients at the Hereditary Cancer Unit of La Paz University Hospital, and concluded that a definitive relationship between MUTYH mutations and breast cancer was not established, emphasizing the need for cautious interpretation in assessing MUTYH's role in breast cancer risk.
Major criticizes:
1. I cannot comprehend the rationale behind the selection criteria: if a known familial genetic variant in cancers existed, only PCR and sequencing of the amplified fragments were performed. However, if a hereditary syndrome was suspected, Next Generation Sequencing (NGS) was conducted. Why is NGS not employed for all samples?
2. In my understanding, NGS results encompass the entire set of genes. Why, then, are only specific genes included for different suspected diseases? For instance, in the case of Familial Adenomatous Polyposis (FAP), only APC and MUTYH are included. Similarly, for patients with early-onset colorectal cancer, the NGS panel comprises MLH1, MSH2, MSH6, PMS2, EPCAM, 104 POLE, POLD, MUTYH, and APC. Could you provide insight into this selective approach? In essence, it might be beneficial to ascertain the prevalence of mutations, including MUTYH, BRCA1, BRCA2, PTEN, among others, in the entire cohort of 3598 selected patients. Additionally, understanding the ratio differences in MUTYH and BRCA mutations between FAP and non-FAP patients, HBOCS and non-HBOCS patients, and the BRCA mutation ratio differences between MUTYH mutation and non-MUTYH mutation patients could provide more informative and helpful insights.
3. The description in the results section lacks clarity and exhibits a chaotic logic. The structure of the table is not organized logically, and it fails to effectively incorporate necessary information, potentially posing challenges for readers' comprehension. I recommend the author reorganize and rewrite the results section. For instance, consider describing 3.3 first instead of 3.1. Additionally, it would be beneficial to explore relevant articles to improve the presentation of results, including the use of tables.
Comments on the Quality of English Language
Grammar issues:
Line 24-25: In the sentence "Nevertheless, the prevalence was not significantly different in the subpopulation with breast cancer," you might consider rephrasing it for clarity: "However, there was no significant difference in the prevalence within the subpopulation with breast cancer."
Line 69: “BRCA1 and BRCA2 are two suppressor tumor genes,” for clarity and grammatical accuracy, you might consider a slight adjustment: "BRCA1 and BRCA2 are two tumor suppressor genes."
Line 71-72: In the sentence:” BRCA has also been related to the repair processes of the oxidative lesion 8-oxoG, where MUTYH also exercises its function.” you may consider the following for a slight improvement in grammar: "BRCA has also been linked to the repair processes of the oxidative lesion 8-oxoG, where MUTYH also performs its function."
Line 85-87:” A retrospective study was carried out from June 2018 to June 2023 at “Hereditary Cancer Unit” of La Paz University Hospital (Madrid). The adult population (>18 years old) that was attended during this period was included.” Here's a slight refinement for improved flow: "A retrospective study was conducted at the 'Hereditary Cancer Unit' of La Paz University Hospital (Madrid) from June 2018 to June 2023. The study included the adult population (>18 years old) attended during this period."
Reviewer 3 Report
Comments and Suggestions for Authors
In this manuscript the authors performed a retrospective study with 3598 patients to investigate the correlation between MUTYH mutation and breast cancer. Authors found MUTYH mutations significantly correlates with colonic polyposis but not breast cancer. This study tried to elucidate the relation between MUTYH mutation and breast cancer, which could brings the implications for diagnosis and screening. Although there were conflict results in the literature this study sided with most of the previous studies stated there were no association. The layout content is short and concise. The conclusion is well supported by the results. The authors fairly presented the strength and the limitations of this study. I would recommend publish this manuscript if the authors could address some minor concerns.
(1) Line 30, breast should be replaced with breast cancer in Keywords.
(2) Line 117, there was a abundant symbol before chi-square tests.
(3) For 3.1 MUTYH variants section, if authors could summarize them into a table that would be more informative.
Round 2
Reviewer 2 Report
Comments and Suggestions for Authors
I appreciate the effort you and your co-authors have put into addressing the reviewers' comments and improving the overall quality of the manuscript. Your revisions have significantly strengthened the content.